# Postoperative joint pain is associated with long-term all-cause mortality after total joint arthroplasty

Ming Liu[1], Andrew Furey[2], Proton Rahman[3], Guangju Zhai [1]*

1 Human Genetics & Genomics, Division of Biomedical Sciences, Faculty of Medicine, Memorial University of Newfoundland, St. John's, Newfoundland and Labrador, Canada, 2 Discipline of Surgery, Faculty of Medicine, Memorial University of Newfoundland, St. John's, Newfoundland and Labrador, Canada, 3 Discipline of Medicine, Faculty of Medicine, Memorial University of Newfoundland, St. John's, Newfoundland and Labrador, Canada

* gzhai@mun.ca

## Abstract

The aims of this study were to assess (1) if postoperative joint pain can predict long-term all-cause mortality after total joint arthroplasty (TJA), and (2) if postoperative joint pain was associated with causes of death (COD) in TJA patients. Patients who underwent total knee or hip arthroplasty were assessed once for their postoperative joint pain at least one-year after TJA using the Western Ontario and McMaster Universities Osteoarthritis Index Likert 3.0 pain subscale. Three pain definitions were utilized: "sustained pain" – pain on all five questions, "pain while active" – pain while walking and taking stairs, and "pain at rest" – pain while sitting/lying and at night while in bed. Patients reporting no pain were classified as controls. Associations between postoperative joint pain and mortality were assessed using Kaplan-Meier survival analysis and multivariable Cox proportional hazards regression to adjust for age at TJA, sex, body mass index (BMI), cardiovascular diseases (CVD), and cancer. The distribution of COD between pain groups and controls were compared using Fisher's exact test. A total of 727 patients were included in the study, of which 129 (18%) were deceased. The prevalence of postoperative sustained pain, pain while active, and pain at rest at 4-year after TJA was 10, 17, and 12%, respectively. The all-cause mortality rate at 11-year after TJA was 20, 26, 19, and 15% in these pain groups and controls, respectively, significantly higher in pain while active group (p = 0.006). Pain while active was positively associated with mortality when knee and hip patients were analyzed together and separately (p ≤ 0.03, hazard ratio (HR) ≥ 1.80), and the significances became stronger after adjusting for age at surgery, sex, BMI, CVD, and cancer (p < 0.001, HR ≥ 2.57). No association was observed between postoperative joint pain and COD. Our results demonstrated that postoperative joint pain could be an important predictor for long-term all-cause mortality in TJA patients.

**Data availability statement:** Data cannot be shared publicly because of the study's ethics application approved by the Health Research Ethics Authority (HREA) of Newfoundland and Labrador (Email: info@hrea.ca). Data are available from the HREA of Newfoundland and Labrador (contact via info@hrea.ca) or corresponding author Dr. Guangju Zhai (contact via gzhai@mun.ca) for researchers who meet the criteria for access to confidential data.

**Funding:** Author GZ received the following funding: the Canadian Institutes of Health Research (https://cihr-irsc.gc.ca/e/193.html, FRN#175015; 153298; 143058; 132178; 191966), the Arthritis Society (https://arthritis.ca/, ID# 22-121), the Research and Development Corporation of Newfoundland and Labrador (https://www.gov.nl.ca/iet/royalties/innovation-business-dev-fund/, 5404.1423.102), and Memorial University of Newfoundland Medical Research Fund (https://www.mun.ca/medicine/research/medical-research-foundation/). The funders had no role in the study design, data collection and analysis, decision to publish, or preparation of the manuscript.

**Competing interests:** The authors have declared that no competing interests exist.

**Abbreviations:** BMI, body mass index; COD, causes of death; CVD, cardiovascular diseases; HR, hazard ratio; NFOAS, Newfoundland Osteoarthritis Study; NL, Newfoundland and Labrador; OA, osteoarthritis; TJA, total joint arthroplasty; WOMAC, Western Ontario and McMaster Universities Osteoarthritis Index

## Introduction

Total joint arthroplasty (TJA) is an effective treatment for end-stage joint diseases to relieve joint pain, restore joint function, and improve patients' quality of life. Over 1.3 million TJAs are performed annually in the United States alone [1], and this number continues to grow as the population ages. The estimated increase of annual arthroplasty volume is projected to be 469–659% by 2060 [2].

Mortality is an often-overlooked outcome measure of elective procedures such as TJA, but risk of death is paramount to patients and their families as they weigh pros and cons of proceeding with TJA. It is important to identify indicators for mortality to better inform patients, enhance patient care algorithms, and monitor surgical care systems. Studies on mortality of TJA patients mainly focus on the utilization of patients' demographic and surgical factors, comorbidities, and revision status in predicting short-term mortality [3–5]. Patients' post-surgery responses are important measures to assess the effectiveness of treatment and quality of recovery. Patient-reported outcome provides valuable information on aspects of patients' health status that are relevant to their quality of life, yet very few studies investigated the associations between these factors and long-term post-surgery mortality in TJA patients.

Despite being a procedure with relatively high success rate, 10–34% of patients still suffer from persistent joint pain after TJA [6]. Postoperative pain is routinely assessed after TJA surgeries to evaluate the effectiveness of the procedure and patient satisfaction towards the surgery [7], however, this information has yet to be utilized in areas beyond monitoring patient recovery. Pain perception is highly subjective. Biological, psychological, and social factors all contribute to the individual variations in pain sensitivity, threshold, and tolerance [8,9]. Meanwhile, these factors could also affect the severity of pain-related conditions, some of which have high prevalence and fatality rates in older patients. It is meaningful to study the associations between pain intensity from the patient's perspective and mortality in TJA patients.

Therefore, we undertook this study to investigate (1) if patient-reported postoperative joint pain can predict long-term all-cause mortality after TJA, and (2) if patient-reported postoperative joint pain was associated with causes of death (COD) in TJA patients.

## Methods

### Study participants

The study participants were derived from the Newfoundland Osteoarthritis Study (NFOAS) which recruited 1086 patients aged at least 20 years old who underwent total knee or hip arthroplasty largely due to osteoarthritis (OA) with a small number of patients due to other joint diseases between 30/11/2011 and 05/02/2017 in St. Clare's Mercy Hospital and Health Science Centre General Hospital in St. John's, Newfoundland and Labrador (NL), Canada [10]. Patients were recruited consecutively. The NFOAS was approved by the Health Research Ethics Authority of Newfoundland and Labrador (HREB # 2011.311) and informed written consent was obtained from all participants.

## Postoperative joint pain assessment

Patients' postoperative pain in the replaced joint was assessed once at least one year after TJA using the Western Ontario and McMaster Universities Osteoarthritis Index (WOMAC) Likert 3.0 pain subscale. This pain subscale consists of five questions evaluating patients' self-reported pain intensity while walking on a flat surface, going up and down stairs, in bed at night, sitting/lying, and standing upright, respectively. Each question is scored on a Likert scale of 0–4, with 0 representing no pain and 4 representing severe pain. For patients who answered at least one but not all five questions, the missing scores were imputed using the mean scores of the given questions. Three pain definitions were utilized in the analyses: "sustained pain" – pain scores ≥ 1 for all five questions; "pain while active" – pain scores ≥ 1 while walking on a flat surface and going up and down stairs; and "pain at rest" – pain scores ≥ 1 while sitting/lying and in bed at night. Patients reporting no pain for all five questions were classified as controls. These case definitions were used previously [11]. For patients who had both knee and hip replaced and had WOMAC pain data for both joints, both records were included in the analysis.

## Demographic and mortality data collection

Sex, date of birth, and comorbidity data were collected using a general health questionnaire prior to the TJA and age at surgery and age at postoperative pain assessment were then calculated. Weight and height data were retrieved from the Eastern Health Meditech Health Care Information System, body mass index (BMI) was then calculated as weight in kg divided by squared height in meters. Data on three mortality variables, patients' mortality, date of death, and COD, were retrieved from the Newfoundland and Labrador Centre for Health Information Mortality System.

## Data analyses

Normality of data distribution was tested with the Shapiro-Wilk test. Age at surgery, age at pain assessment, and BMI, which were not normally distributed, were compared using the Mann-Whitney $U$ test. Sex distribution, prevalence of 37 comorbidities, and percentage of patients with multiple comorbidities (number of comorbidities > 2 to > 10) were tested with *Chi*-squared test or Fisher's exact test. The associations between postoperative joint pain and mortality were assessed using Kaplan-Meier survival analysis, and multivariable Cox proportional hazards regression was utilized to adjust for age at surgery, sex, BMI, cardiovascular diseases (CVD), and cancer [12]. COD between pain groups and controls were compared using Fisher's exact test. Analyses were performed for combined knee and hip joints as well as joint specifically. Significance level for prevalence of comorbidities/percentage of patients with multiple comorbidities was set at $\alpha = 0.001$ to control for multiple testing of 46 comparisons using the Bonferroni method, and that for other variables was set at $\alpha = 0.05$. All analyses were performed in R version 4.2.3 with survival [13] and survminer [14] packages. Visualizations of the results were performed with ggplot2 R package [15].

## Results

Among the 1,086 consented patients, 95 patients had bipolar hemiarthroplasty, and TJA for three patients was cancelled, hence these patients were excluded from the analysis. Mortality data for 26 patients could not be linked because they lived outside of NL/Canada. Among the remaining 962 patients, 38 deceased prior to postoperative joint pain assessment, 30 were unable to participate due to health reasons, 27 withdrew consent or unwilling to answer questions, and 140 were unreachable (Fig 1). Out of the remaining 727 patients, four had both knee and hip replaced, therefore a total of 731 records were included in the final analyses (Fig 1). Compared to these patients, the excluded patients had higher age and lower BMI ($p < 0.001$, S1 Table).

For combined knee and hip patients, 55% were women; mean age was 64.8 years and mean BMI was 33.6 kg/m². There were 493 patients who had knee arthroplasty, of which 57% were women; mean age was 65.0 years and mean BMI

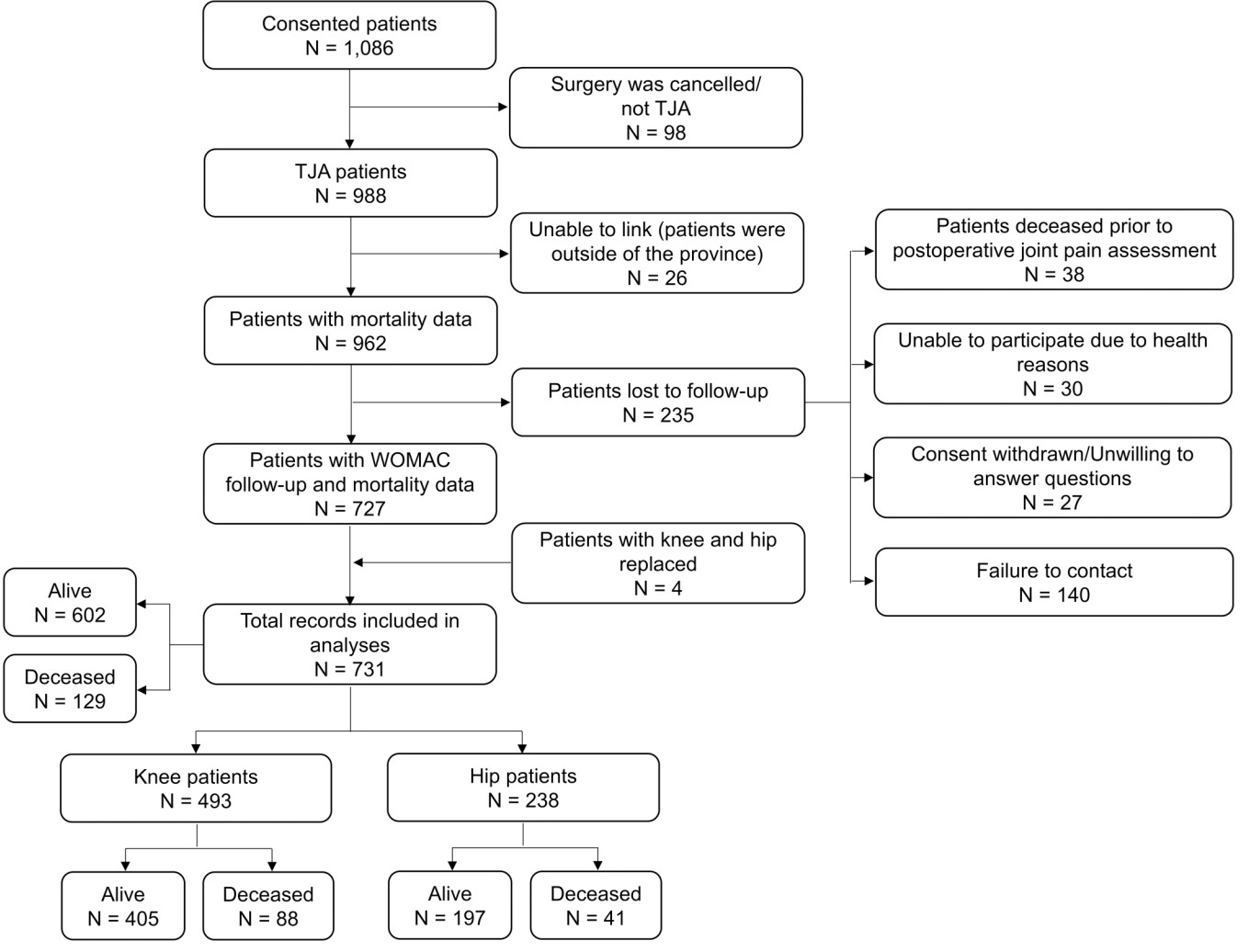

**Fig 1. Flowchart of study participants.** WOMAC: Western Ontario and McMaster Universities Osteoarthritis Index.

was 33.6 kg/m². There were 238 patients who had hip arthroplasty, of which 50% were women; mean age was 64.5 years and mean BMI was 31.3 kg/m². Hip patients had significantly lower BMI compared to knee patients ($p < 0.001$) but there was no difference in age or sex distribution ($p = 0.99$ and 0.06, respectively). More details are presented in Table 1.

### Postoperative joint pain and mortality

The average follow-up time for postoperative pain in the operated joint was 4.0 years, and the last follow-up date was July 29, 2019. Out of the 731 WOMAC records, 725 had complete scores, and imputation was performed for the other six records that each had one or two missing scores. Mortality data was collected 11.0 years on average after the TJA. Data on mortality, date of death, and COD for patients who deceased prior to January 2022 were extracted on June 3, 2024, and COD data for patients who deceased in 2022 were extracted on January 15, 2025. The COD data for patients who deceased after 2022 were unavailable due to the timeline of data release to the Newfoundland and Labrador Centre for Health Information Mortality System.

**Table 1. Characteristics of the study participants #.**

| Knee + Hip | All (n = 731) | Control (n = 546, 75%) | Sustained pain (n = 70, 10%) | P-value | Pain while active (n = 124, 17%) | P-value | Pain at rest (n = 89, 12%) | P-value |
|---|---|---|---|---|---|---|---|---|
| Female (%) | 55 | 54 | 57 | 0.62 | 56 | 0.62 | 55 | 0.86 |
| Age at surgery (yrs) | 64.81±8.77 | 65.69±8.62 | 59.36±9.02 | <0.001** | 61.76±8.93 | <0.001** | 60.00±8.87 | <0.001** |
| Age at pain assessment (yrs) | 68.80±8.76 | 69.73±8.59 | 63.16±9.13 | <0.001** | 65.48±8.98 | <0.001** | 63.79±8.90 | <0.001** |
| BMI (kg/m²) | 33.57±6.95 | 33.39±7.06 | 33.10±6.18 | 0.83 | 33.98±6.59 | 0.21 | 33.13±6.14 | 0.77 |
| **Knee** | **All (n = 493)** | **Control (n = 361, 73%)** | **Sustained pain (n = 44, 9%)** | **P-value** | **Pain while active (n = 89, 18%)** | **P-value** | **Pain at rest (n = 57, 12%)** | **P-value** |
| Female (%) | 57 | 56 | 61 | 0.52 | 60 | 0.57 | 56 | 0.99 |
| Age at surgery (yrs) | 64.97 ± 7.95 | 65.64±8.02 | 60.05±7.47 | <0.001** | 62.57±8.02 | 0.003* | 60.83±7.49 | <0.001** |
| Age at pain assessment (yrs) | 68.97±7.96 | 69.73±8.02 | 63.80±7.31 | <0.001** | 66.21±7.98 | <0.001** | 64.59±7.39 | <0.001** |
| BMI (kg/m²) | 33.64±6.91 | 34.43±7.04 | 34.71±6.53 | 0.53 | 35.34±6.63 | 0.14 | 34.67±6.24 | 0.48 |
| **Hip** | **All (n = 238)** | **Control (n = 185, 78%)** | **Sustained pain (n = 26, 11%)** | **P-value** | **Pain while active (n = 35, 15%)** | **P-value** | **Pain at rest (n = 32, 13%)** | **P-value** |
| Female (%) | 50 | 50 | 50 | 0.98 | 49 | 0.90 | 53 | 0.72 |
| Age at surgery (yrs) | 64.49±10.27 | 65.79±9.70 | 58.21±11.24 | <0.001** | 59.68±10.76 | <0.001** | 58.52±10.88 | <0.001** |
| Age at pain assessment (yrs) | 64.45±10.23 | 69.73±9.64 | 62.07±11.66 | <0.001** | 63.64±11.05 | 0.001* | 62.35±11.08 | <0.001** |
| BMI (kg/m²) | 31.34±6.50 | 31.33±6.63 | 30.38±4.45 | 0.97 | 30.52±5.09 | 0.77 | 30.38±4.94 | 0.89 |

#: Values are the mean ± standard deviation unless indicated otherwise. *P*-values were obtained by the Mann-Whitney *U* test, Student's t-test, or *Chi*-squared test wherever appropriate comparing each pain group with controls; *: *P*-value < 0.05; **: *P*-value < 0.001. BMI: body mass index.

For combined knee and hip patients, prevalence of postoperative sustained pain, pain while active, and pain at rest was 10, 17, and 12%, respectively, and 75% of patients were controls. Pain patients were significantly younger than controls at the time of surgery as well as pain assessment ($p < 0.001$, Table 1). Prevalence of clinical depression, Crohn's disease, migraine, golfer's elbow, and polycystic ovary syndrome and percentage of patients with more than 8 comorbidities were higher in pain groups, but the differences were not significant after controlling for multiple testing ($p \le 0.04$, S2 Table). The mortality rate was 18% for the entire cohort, and 20, 26, 19, and 15% for the three pain groups and controls, respectively, significantly higher in pain while active group compared to controls ($p = 0.006$, Table 2). Deceased patients were older at the time of TJA and included more men compared with those who were alive ($p \le 0.003$). The demographic factors of deceased and alive patients are presented in Table 3.

For knee joint, prevalence of postoperative sustained pain, pain while active, and pain at rest was 9, 18, and 12%, respectively, and 73% of patients were controls (Table 1). The mortality rate was 18% for all knee patients, and 18, 25, 19, and 16% for the three pain groups and controls, respectively, not significantly different between pain groups and controls ($p \ge 0.06$, Table 2).

**Table 2. All-cause mortality rate after joint arthroplasty in control and pain groups #.**

| | Control | | Sustained pain | | | Pain while active | | | Pain at rest | | |
|---|---|---|---|---|---|---|---|---|---|---|---|
| Joint Group | N | Mortality rate (%) | N | Mortality rate (%) | P-value | N | Mortality rate (%) | P-value | N | Mortality rate (%) | P-value |
| **Knee + Hip** | 546 | 15 | 70 | 20 | 0.32 | 124 | 26 | 0.006* | 89 | 19 | 0.37 |
| **Knee** | 361 | 16 | 44 | 18 | 0.72 | 89 | 25 | 0.06 | 57 | 19 | 0.54 |
| **Hip** | 185 | 14 | 26 | 23 | 0.23 | 35 | 29 | 0.03* | 32 | 19 | 0.49 |

#: *P*-values were obtained by the *Chi*-squared test comparing each pain group with controls; *: *P*-value < 0.05.

**Table 3. Patient characteristics by death status in control and pain groups #.**

| Knee+Hip | Control (n=546) | | | Sustained pain (n=70) | | | Pain while active (n=124) | | | Pain at rest (n=89) | | |
|---|---|---|---|---|---|---|---|---|---|---|---|---|
| | Deceased (n=84) | Alive (n=462) | P-value | Deceased (n=14) | Alive (n=56) | P-value | Deceased (n=32) | Alive (n=92) | P-value | Deceased (n=17) | Alive (n=72) | P-value |
| Female (%) | 39 | 57 | 0.003* | 64 | 55 | 0.55 | 53 | 58 | 0.66 | 65 | 53 | 0.37 |
| Age at surgery (yrs) | 71.25±8.57 | 64.68±8.24 | <0.001** | 65.06±6.76 | 57.94±8.99 | 0.01* | 66.77±7.62 | 60.01±8.73 | <0.001** | 63.88±6.87 | 59.08±9.08 | 0.09 |
| BMI (kg/m²) | 32.00±6.49 | 33.63±7.13 | 0.04* | 32.43±3.99 | 33.27±6.64 | 0.95 | 33.69±5.74 | 34.08±6.88 | 0.97 | 33.06±4.15 | 33.14±6.54 | 0.63 |

| Knee | Control (n=361) | | | Sustained pain (n=44) | | | Pain while active (n=89) | | | Pain at rest (n=57) | | |
|---|---|---|---|---|---|---|---|---|---|---|---|---|
| | Deceased (n=58) | Alive (n=303) | P-value | Deceased (n=8) | Alive (n=36) | P-value | Deceased (n=22) | Alive (n=67) | P-value | Deceased (n=11) | Alive (n=46) | P-value |
| Female (%) | 40 | 59 | 0.005* | 63 | 61 | 1.00 | 50 | 63 | 0.29 | 64 | 54 | 0.58 |
| Age at surgery (yrs) | 71.16±8.37 | 64.58±7.51 | <0.001** | 65.87±3.55 | 58.75±7.52 | 0.009* | 67.69±6.41 | 60.89±7.82 | <0.001** | 63.82±5.11 | 60.11±7.83 | 0.22 |
| BMI (kg/m²) | 33.04±6.51 | 34.70±7.12 | 0.12 | 31.98±2.91 | 35.32±6.98 | 0.25 | 33.98±5.04 | 34.79±7.05 | 0.31 | 33.09±3.64 | 35.05±6.69 | 0.45 |

| Hip | Control (n=185) | | | Sustained pain (n=26) | | | Pain while active (n=35) | | | Pain at rest (n=32) | | |
|---|---|---|---|---|---|---|---|---|---|---|---|---|
| | Deceased (n=26) | Alive (n=159) | P-value | Deceased (n=6) | Alive (n=20) | P-value | Deceased (n=10) | Alive (n=25) | P-value | Deceased (n=6) | Alive (n=26) | P-value |
| Female (%) | 38 | 52 | 0.22 | 67 | 45 | 0.64 | 60 | 44 | 0.47 | 67 | 50 | 0.66 |
| Age at surgery (yrs) | 71.44±9.16 | 64.87±9.50 | 0.005* | 63.98±9.94 | 56.47±11.25 | 0.22 | 64.75±9.87 | 57.66±10.61 | 0.15 | 63.98±9.94 | 57.25±10.84 | 0.17 |
| BMI (kg/m²) | 29.47±5.80 | 31.61±6.72 | 0.07 | 33.03±5.36 | 29.59±3.95 | 0.04* | 33.06±7.33 | 29.51±3.59 | 0.15 | 33.03±5.36 | 29.77±4.74 | 0.07 |

#. Values are the mean ± standard deviation unless indicated otherwise. $P$-values were obtained by the Mann–Whitney $U$ test, $Chi$-squared test, or Fisher's exact test wherever appropriate comparing deceased and alive patients in each group; *: $P$-value <0.05; **: $P$-value <0.001. BMI: body mass index.

For hip joint, prevalence of postoperative sustained pain, pain while active, and pain at rest was 11, 15, and 13%, respectively, and 78% of patients were controls (Table 1). The mortality rate was 17% for all hip patients, and 23, 29, 19, and 14% for the three pain groups and controls, respectively, significantly higher in pain while active group compared to controls ($p = 0.03$, Table 2).

Kaplan-Meier survival analysis showed that pain while active was positively associated with mortality with a hazard ratio (HR) of 1.92, 1.80, and 2.18 for knee and hip combined, knee, and hip, respectively ($p \leq 0.03$, Table 4), and the HR became higher after adjusting for age at surgery, sex, BMI, CVD, and cancer (HR = 2.96, 2.57, and 4.31 for knee and hip combined, knee, and hip, respectively, $p < 0.001$, Fig 2). Sustained pain or pain at rest was not associated with mortality ($p \geq 0.25$, Table 4).

### Postoperative joint pain and COD

COD data were available for 78 patients and classified into eight categories: neurological conditions, cardiac/circulatory conditions, pulmonary conditions, digestive system conditions, genito/urinary/reproductive conditions, infections, neoplasms, and unspecified conditions. The two leading CODs were cardiac/circulatory conditions and neoplasms, causing 35 and 27% of deaths, respectively (Table 5). Cardiac/circulatory conditions were the leading COD in the three pain groups and controls, responsible for 83, 40, 56, and 34% of deaths, respectively. Neoplasms were the second leading COD in all four groups, responsible for 17, 30, 22, and 24% of deaths, respectively (Table 5). There was no difference in COD between pain groups and controls ($p \geq 0.47$, Table 5).

### Discussion

Assessing long-term mortality following TJA is important to help elucidate the potential benefits of TJA to overall patient health. Meta-analysis on studies published between 2011 and 2021 showed that the average 10-year mortality rate was 16.43% after total hip arthroplasty [16] and 10.18% after total knee arthroplasty [17]. Our cohort had a comparable mortality rate in hip patients (17%) but higher rate in knee patients (18%). This could be due to the differences in study populations and study duration. Our NL population has a unique genetic structure [18] which could affect its susceptibility to diseases, and our evaluation for mortality covered the entire coronavirus disease 2019 (COVID-19) pandemic period [19] during which higher mortality rate was observed in TJA patients [20].

Older age and male sex have been reported to be risk factors for long-term mortality after TJA [21,22]. We also observed deceased patients were most likely to be men and older at the time of surgery than the patients who were alive. Older age has long been recognized as a risk factor for post-surgery mortality which has been connected to the increased vulnerability to stressors resulting in decreased physiological reserves and deregulation of multiple systems [23]. The higher percentage of men in the deceased group could be due to the lower life expectancy of men, which has been attributed to the differences in biological characteristics such as sex hormones and genetic factors, stress level, as well as behavioral and epidemiological factors between men and women [24,25].

**Table 4. Kaplan-Meier survival estimates #.**

|  | Sustained pain | | Pain while active | | Pain at rest | |
|---|---|---|---|---|---|---|
| Joint Group | HR (95% CI) | *P*-value | HR (95% CI) | *P*-value | HR (95% CI) | *P*-value |
| **Knee + Hip** | 1.38 (0.78, 2.43) | 0.26 | 1.92 (1.28, 2.88) | 0.001* | 1.36 (0.81, 2.28) | 0.25 |
| **Knee** | 1.22 (0.58, 2.56) | 0.60 | 1.80 (1.10, 2.94) | 0.02* | 1.34 (0.70, 2.55) | 0.37 |
| **Hip** | 1.67 (0.69, 4.06) | 0.25 | 2.18 (1.05, 4.52) | 0.03* | 1.38 (0.57, 3.35) | 0.48 |

#: *P*-values, hazard ratios, and confidence intervals were obtained by the Kaplan-Meier survival analysis; *: *P*-value < 0.05. HR: hazard ratio; CI: confidence interval.

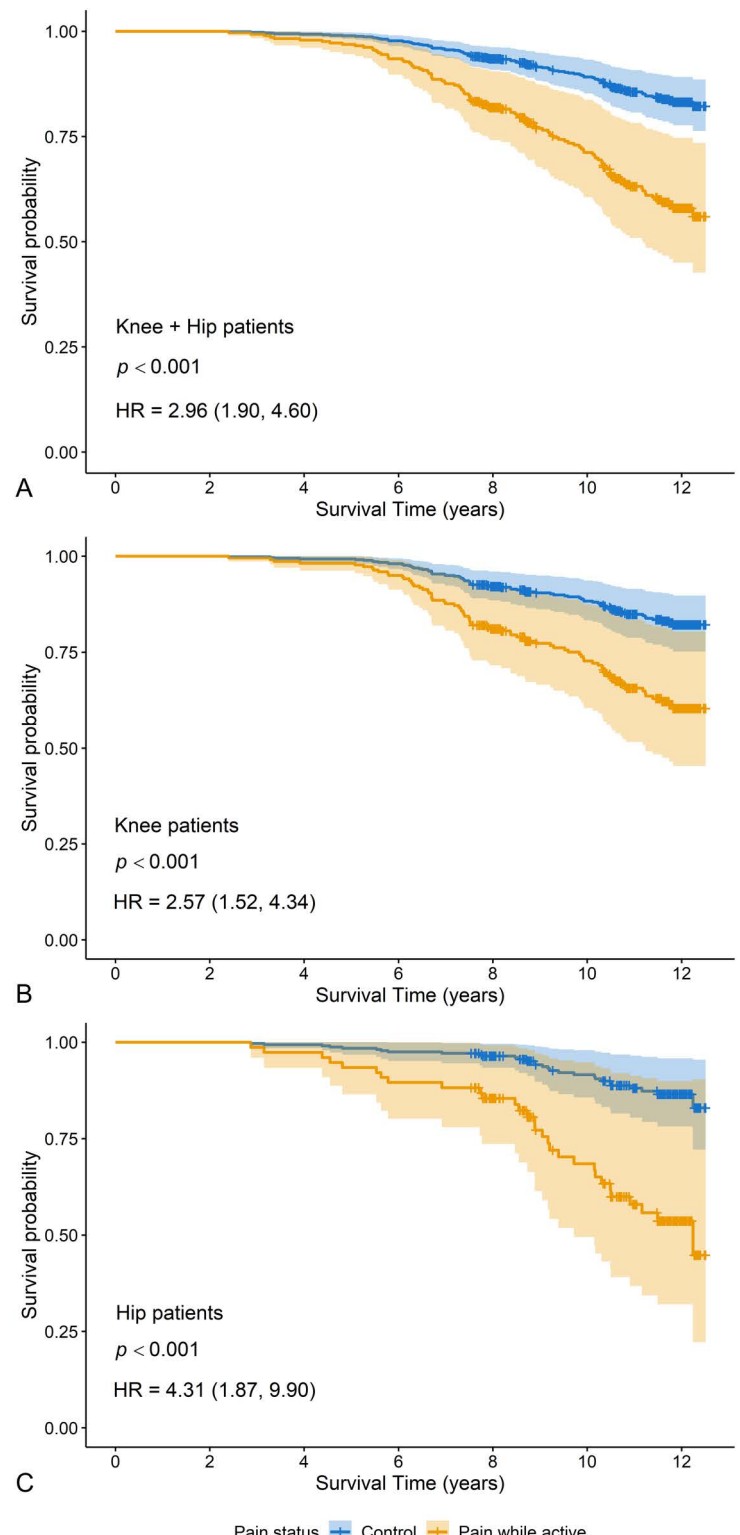

**Fig 2. Cox survival plot for pain while active in knee and hip patients combined (A), knee patients (B) and hip patients (C).** *P*-values, hazard ratios, and confidence intervals were obtained by the multivariable Cox proportional hazards regression analysis adjusting for age, sex, body mass index, cardiovascular diseases, and cancer. HR: hazard ratio.

**Table 5. Cause of death in study groups.**

| COD | All with COD (n=78) N (%) | Controls with COD (n=50) N (%) | Sustained pain with COD (n=8) N (%) | P-Value * | Pain while active with COD (n=22) N (%) | P-Value * | Pain at rest with COD (n=9) N (%) | P-Value * |
|---|---|---|---|---|---|---|---|---|
| | | | | 0.65 | | 0.90 | | 0.47 |
| Cardiac/circulatory conditions | 27 (35) | 17 (34) | 5 (83) | | 8 (40) | | 5 (56) | |
| Neoplasms | 21 (27) | 12 (24) | 1 (17) | | 6 (30) | | 2 (22) | |
| Pulmonary conditions | 13 (17) | 10 (20) | – | | 2 (10) | | – | |
| Neurological conditions | 6 (8) | 3 (6) | – | | 2 (10) | | 1 (11) | |
| Infections | 5 (6) | 3 (6) | – | | 1 (5) | | – | |
| Digestive system conditions | 2 (3) | 1 (2) | – | | 1 (5) | | 1 (11) | |
| Genito/urinary/reproductive conditions | 2 (3) | 2 (4) | – | | – | | – | |
| Unspecified | 2 (3) | 2 (4) | – | | – | | – | |

* *P*-values were obtained by the Fisher's exact test comparing each pain group with controls. COD: cause of death.

## Postoperative joint pain and mortality

WOMAC is a widely used self-administered questionnaire to evaluate patient-reported joint pain [26], and it has been demonstrated that the WOMAC pain subscale has a high test-retest reliability [27]. Using WOMAC pain subscale and the three pain definitions, we observed that 9–18% patients experienced chronic joint pain after TJA, consistent with previous reports [6]. Our results showed that pain while active was positively associated with mortality at 11 years after TJA, and this association was not specific to patients who had knee or hip arthroplasty, suggesting that this type of pain might affect patients on a broader scope. To date, the only other study that investigated the predictive value of postoperative WOMAC pain scores in long-term mortality of TJA patients was a study conducted by Klimek et al. which reported no association between joint-specific WOMAC assessment and long-term mortality [28]. The conflicting findings could be due to the differences in pain evaluation period, pain classification method, as well as study population as mentioned above. Klimek et al. evaluated joint pain at 12-month after TJA in a German cohort, while our pain evaluation was conducted at four years on average after TJA in the NL population, when the pain had become more chronic. The pain intensity in their study was classified into four groups based on the total pain scores, 0–8, 9–11, 12–14, and 15–20, while our study used three pain definitions based on the characteristics of pain reflected by the five questions, which were more relevant to the impairment of different aspects of patients' daily activities [11]. Our findings are supported by previous reports which showed that joint pain was associated with 21–65% increased risks of death [29], and chronic pain nearly doubled mortality rate [30].

We did not observe any association between sustained pain or pain at rest and mortality after TJA, which could be due to the smaller sample sizes in these pain groups. Future studies are needed to further investigate the predictive values of these two types of pain in mortality in TJA patients.

## Postoperative joint pain and COD

We also examined the relationships between postoperative joint pain and COD in TJA patients but did not find any significant association. The conditions that caused most of the deaths in pain patients and controls were cardiac/circulatory conditions followed by neoplasms. These two conditions were previously reported as leading CODs in TJA patients [17]. This indicated that postoperative joint pain might accelerate the courses of these diseases and eventually lead to higher mortality.

Pain while active is negatively associated with total volume of physical activity mediated by physical activity enjoyment [31]. Physical activity has been shown to have inverse relationships with all-cause mortality and deaths from cardiovascular diseases, coronary heart diseases, and cancers [32,33], all of which were leading CODs in patients with postoperative pain in our study. It is plausible that the positive association between pain while active and mortality observed in the current study might be driven by deaths from these conditions in which lower levels of physical activity played a role. In addition, conditions causing joint pain are associated with an increased prevalence of long-term opioid use [34]. Opioids have been widely used for postoperative pain management and 24–29% of senior patients reported being on opioids at 12-month after TJA [35,36]. Chronic opioid use is associated with increased risk for myocardial infarction/coronary revascularization and can cause a myriad of cardiovascular complications [37]. Unfortunately, we did not have data on opioid use in our cohort. The role of opioid use in the relationship between postoperative pain and mortality requires further investigation. Another common observation in chronic pain patients is the high prevalence of depression. In our cohort, 11% of patients with pain while active reported having clinical depression while the prevalence in control group was only 4% (S2 Table). Evidence has suggested that chronic pain and depression may be based on common neuroplasticity mechanism changes and their coexistence tends to further aggravate the severity of both disorders [38]. Depression is positively associated with cancer recurrence and mortality [39], and can also lead to an increased risk of cardiovascular disease by driving the development or exacerbation of traditional cardiac risk factors [40]. Interestingly, cardiac rehabilitation and exercise training has been shown to reduce psychological distress, especially depression [40], while pain is limiting patients in performing these exercises. The intertwined relationships between chronic pain, depression, and cardiovascular diseases as well as cancers could be important players behind the observed high mortality in patients with postoperative joint pain.

Alleviating joint pain or helping patients to achieve a pain-free state is the primary goal of TJA and the focus in postoperative care. However, the observation of the current study raised an interesting question: are all joint pains equal? Or, in other words, is treating pain itself enough in patients with joint pain during activities after TJA? Given the previous evidence on the associations between pain, physical activity, depression, and cardiovascular diseases and cancers, it might be time to open new topics for discussion: whether treating other conditions closely related to pain and helping these patients resume regular daily activities should also be important parts in the therapeutic regime for TJA patients, a question not only involving the health care system but also social support.

There were some limitations in this study. The percentage of study participants lost to follow-up was high as often seen in long-term follow-up studies, and age and BMI differed between the included and excluded participants, and this could potentially bias the results. Data on COD was not available for all patients, which further reduced our sample size and statistical power. Post-operative joint pain scores were imputed for a small number of patients, although it was unlikely that this would have affected the results given that > 99% of the WOMAC records were complete. As the main aim of this study was to investigate whether postoperative joint pain as the outcome of TJA could be associated with long-term mortality, our analysis did not account for preoperative level of pain. Studies on the relationship between surgery response such as the changes between preoperative and postoperative joint pain and mortality are also needed to broaden our understanding of the impact of TJA on mortality. In addition, no data on medication use was available, so we were unable to determine whether patients in the control group were indeed pain free or taking pain medication for other comorbidities. Furthermore, the unique structure of the NL population could limit the generalizability of our findings to other populations. More studies are needed to further evaluate the predictive value of postoperative joint pain in long-term mortality and investigate the deeper connections between such pain and mortality in TJA patients.

## Conclusions

Patient-reported postoperative joint pain, particularly pain while active, could be a valuable predictor of long-term all-cause mortality in TJA patients. The findings could help inform patients in postoperative care and develop more targeted patient care algorithms in TJA patients.

## Supporting information

**S1 Table. Patient characteristics of included and excluded TJA patients.** Values are the mean ± standard deviation unless indicated otherwise. *P*-values were obtained by the Mann-Whitney *U* test or *Chi*-squared test wherever. *: *P*-value < 0.001. BMI: body mass index; TJA: total joint arthroplasty.
(DOCX)

**S2 Table. Comparison of prevalence of comorbidities and percentage of patients with multiple comorbidities between pain groups and controls.** Values are *p* values obtained with *Chi*-squared test or Fisher's exact test comparing each pain group with controls. Values smaller than 0.05 are marked in bold.
(DOCX)

## Acknowledgments

We thank all study participants who made this study possible.

## Author contributions

**Conceptualization:** Guangju Zhai.

**Data curation:** Ming Liu, Andrew Furey, Proton Rahman, Guangju Zhai.

**Formal analysis:** Ming Liu, Guangju Zhai.

**Funding acquisition:** Andrew Furey, Proton Rahman, Guangju Zhai.

**Investigation:** Ming Liu, Guangju Zhai.

**Methodology:** Guangju Zhai.

**Resources:** Andrew Furey, Proton Rahman, Guangju Zhai.

**Supervision:** Guangju Zhai.

**Visualization:** Ming Liu.

**Writing – original draft:** Ming Liu, Guangju Zhai.

**Writing – review & editing:** Ming Liu, Andrew Furey, Proton Rahman, Guangju Zhai.

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
