## [Decision Letter · Decision Letter 0]

PONE-D-25-06222Postoperative joint pain is associated with long-term all-cause mortality after total joint arthroplastyPLOS ONE

Dear Dr. Zhai,

Thank you for submitting your manuscript to PLOS ONE. After careful consideration, we feel that it has merit but does not fully meet PLOS ONE’s publication criteria as it currently stands. Therefore, we invite you to submit a revised version of the manuscript that addresses the points raised during the review process.

We look forward to receiving your revised manuscript.

Kind regards,

Yuanyuan Wang, PhD

Academic Editor

PLOS ONE

Additional Editor Comments:

The reviewers have made comments to improve the manuscript, in particular statistical analysis, interpretation of the results, and implication of study findings.

Reviewers' comments:

Reviewer's Responses to Questions

**Comments to the Author**

1. Is the manuscript technically sound, and do the data support the conclusions?

Reviewer #1: Partly

Reviewer #2: Partly

Reviewer #3: No

2. Has the statistical analysis been performed appropriately and rigorously? 

Reviewer #1: Yes

Reviewer #2: No

Reviewer #3: I Don't Know

3. Have the authors made all data underlying the findings in their manuscript fully available?

Reviewer #1: No

Reviewer #2: No

Reviewer #3: Yes

4. Is the manuscript presented in an intelligible fashion and written in standard English?

Reviewer #1: Yes

Reviewer #2: Yes

Reviewer #3: Yes

5. Review Comments to the Author

Reviewer #1: The manuscript “Postoperative joint pain is associated with long-term all-cause mortality after total joint Arthroplasty” examined the effect of post-operative pain on mortality among patients undergoing TJR. Specific comments and clarifications are below:

1. Abstract (lines 14-16): “… their postoperative joint pain at least one-year after TJA…” Was WOMAC collected at several time points after surgery? Please clarify.

2. Methods, Page 5: Please include the exact duration of the follow-up period (e.g., last date, average # of years of follow-up) and the number of post-operative measurements.

3. Page 5, lines 79-80: Please clarify what post-operative time point was used for the analysis.

4. Page 6: I think that the label ‘sustained pain’ is misleading as it is based on a one-time pain reporting based on a question asking for pain in the past 48 hours.

Furthermore, I think that post-operative pain status is not the best variable to accomplish the study objectives, as it does not account for the level of pre-operative pain. I think a variable looking at changes in pain pre- and post- operatively is more appropriate for the study question. Please comment on the choice of the pain variables used.

5. Page 7: Given that comorbidity data was available, I wonder why the models did not control for comorbidities. Please, clarify

6. Results, Page 7: Given the significant attrition (33%), please expand on the sources of loss to follow-up. For example, how many did not have mortality data and why (e.g., unable to link?), how many did not complete WOMAC? Additionally, how the sample of patients include and excluded differ in terms of basic demographic variables collected. Please include this information in the discussion of the study limitations.

7. Table 1 (knee): the data for age and sex appears to be scrambled.

8. Table 4: Are these estimates from Kaplan-Meier or Cox models?

9. Supplementary Table 1: I suggest including the prevalence of comorbidities. I don’t think that p-values without the prevalence are useful.

Reviewer #2: Authors present retrospective, secondary analysis of data derived from the Newfoundland Osteoarthritis Study regarding correlation between WOMAC pain scores (classified as ‘sustained pain,’ ‘pain while active,’ and ‘pain at rest) and long-term (11-year) mortality with the idea that post-arthroplasty pain is understudied outside of directly assessing surgical outcomes and that this is an important area of study as chronic pain may have important consequences on health outcomes and mortality. Authors set the stage by reviewing the rising use of arthroplasty surgery as well as previously published work discussing mortality and post-operative pain following hip and knee arthroplasty. After analysis (Kaplan-Meier survival analysis, multivariable Cox proportional hazards regression) the authors conclude that “patient-reported postoperative joint pain, particularly pain while active, could be a valuable predictor of long-term all-cause mortality in TJA patients” and further recommend that this pain be addressed in patients who experience it.

The authors (2 from the division of Genetics & Genomics, 1 from the department of internal medicine, 1 from the department of surgery) having educational training in epidemiology and clinical experience caring for this patient population appear to have the appropriate training to undertake this study.

While successful arthroplasty surgery and the prevention of post-operative pain/mortality are important topics, there are several considerations to be addressed in this manuscript before it will add clearly and positively to the literature.

1. Statistical analysis. Do the authors control for any of the comorbidities listed in the primary analysis? It does not appear that this was the case. It would be important to control for comorbidities (coronary artery disease, chronic obstructive pulmonary disease, peripheral arterial disease, diabetes) which are likely stronger predictors of mortality post-operative mortality in the post-operative setting than pain. It would also be important to control for these comorbidities as some of them may be associated with pain with activity (i.e., CAD, PAD). The authors mention 37 comorbidities being examined with Chi-squared or Fisher’s exact test – however many of these comorbidities are not relevant predictors of mortality. Olsen et al. offer a more concise and relevant list of comorbidities in the citation below.

Olsen F, Hård Af Segerstad M, Nellgård B, Houltz E, Ricksten SE. The role of bone cement for the development of intraoperative hypotension and hypoxia and its impact on mortality in hemiarthroplasty for femoral neck fractures. Acta Orthop. 2020 Jun;91(3):293-298. doi: 10.1080/17453674.2020.1745510. Epub 2020 Apr 2. PMID: 32237931; PMCID: PMC8023921.

2. Selection and interpretation of PROM (WOMAC). The authors bin patients into four groups (‘sustained pain’ versus ‘pain while active’ versus ‘pain at rest’ and controls). Is there any basis for using this approach previously in the literature? The authors citation for utilizing this approach references one of their previous studies which correlated metabolic biomarkers with ‘sustained’ pain. This previous work does not adequately prove that parsing apart the WOMAC measure in this way is valid. How specifically were patients binned? If a patient answered a 1 on both activity (walking, going up/down stairs) questions and a 1 on only a single rest (pain at night, pain while sitting) question were they still binned as a “pain while active” patient? What about the contralateral scenario?

3. Discussion/interpretation of results. The conclusion that ‘Efforts should be taken to address such pain in these patients, especially in those with preexisting cardiac/circulatory conditions and/or neoplasms to potentially reduce increased mortality’ highlights the disconnect between the studies methodology, conclusions, and the literature. This study’s findings are too preliminary to make a recommendation such as this. Further, the recommendation is disconnected form the clinical literature as arthroplasty surgeons are always trying to treat their patient’s pain – so they are already doing this. The introduction presents a more compelling interpretation of the goals for the data: ‘It is important to identify indicators for mortality to better inform patients, enhance patient care algorithms, and monitor surgical care systems.’ The discussion could also be made more concise.

Other Specifics:

-70: Is there a citation that can be included that can link to the study protocol or provide more detail on the index study. It is outside the scope of this specific article but readers should be able to easily access what the goal of the original study is, what the inclusion/exclusion criteria were, etc.

-79: Is there a citation that can be included which provides data on the validity of the WOMAC?

-117-118: Not ideal loss to follow-up.

-97-99 + page 16 ‘postoperative joint pain and COD’ section: cause of death data was only available for 10% of the original sample.

Reviewer #3: I thank the authors and the editorial team for the opportunity to review their work on the association of joint pain one year postoperatively and mortality. I think I do understand how this study came to be conceptually, but have difficulties to understand the conclusions drawn from the results. Foremost: What would your biological explanation be for the established association? Would mortality go down if we would give patients more pain killers? Why did your model not include comorbidity even though you had access to this important information? In summary, I am afraid that I do not consider that this study adds to the literature in a meaningful way.

6. PLOS authors have the option to publish the peer review history of their article (what does this mean? ). If published, this will include your full peer review and any attached files.

**Do you want your identity to be public for this peer review?** For information about this choice, including consent withdrawal, please see our Privacy Policy .

Reviewer #1: No

Reviewer #2: No

Reviewer #3: **Yes: ** Anders Brüggemann, MD, PhD

---

## [Author Response · Author response to Decision Letter 1]

28 May 2025

Response:

We have reformatted the manuscript to meet the PLOS ONE's style requirements as instructed.

Response:

ORCID iD for the corresponding author has been validated.

Response:

We have clarified in the Methods section that “informed written consent was obtained from all participants.” (Revised Manuscript with Track Changes, Page 5, lines 81-82)

Response:

The ethical application for this study that was approved by the Health Research Ethics Authority (HREA) of Newfoundland and Labrador specified that all electronic data for this study shall be stored on a secure Health Science Information and Media Service (HSIMS) server with restricted password-protected access as well as password-protected databases, and the data guardian is Dr. Guangju Zhai. The contact information for HREA of Newfoundland and Labrador is as follows: Email: info@hrea.ca, Phone: (709) 864-8871.

Data availability statement:

Data cannot be shared publicly because of the study's ethics application approved by the Health Research Ethics Authority (HREA) of Newfoundland and Labrador (Email: info@hrea.ca). Data are available from the HREA of Newfoundland and Labrador (contact via info@hrea.ca) or corresponding author Dr. Guangju Zhai (contact via gzhai@mun.ca) for researchers who meet the criteria for access to confidential data.

Response:

The ethics statement now only appears in the Methods section, and the statement written in other sections has been deleted (Revised Manuscript with Track Changes, Page 25, Lines 330-335).

Response:

We have added captions for our Supporting Information files at the end of our manuscript (Revised Manuscript with Track Changes, Page 33, Lines 483-492), and updated in-text citations to match accordingly (Revised Manuscript with Track Changes, Page 8, line 132-133; Page 11, line 166, Page 23, line 289).

Additional Editor Comments:

The reviewers have made comments to improve the manuscript, in particular statistical analysis, interpretation of the results, and implication of study findings.

Response:

We thank the reviewers for their constructive comments and have revised the manuscript accordingly.

Query on May 22, 2025:

1. We dispatched the following message on May 20th, 2025. We note that you have resubmitted your manuscript/updated your statement, but have not replied directly to this query. Additionally, your statement and answer remain the same. Please respond to the following query at your earliest convenience.

Your Data Availability statement currently reads:

"Data cannot be shared publicly because of confidentiality reasons."

Additionally, you indicated that your data is not publicly available, and checked the “No - some restrictions will apply" box when submitting your manuscript.

You also provided some Data Availability information via your Response to Reviewers:

"We have stated in “Availability of data and materials” section that “The datasets used and/or analyzed during the current study are available from the corresponding author on upon reasonable request.” (Revised Manuscript with Track Changes, Page 25, Lines 338-339). The ethical application for this study that was approved by the Health Research Ethics Authority (HREA) of Newfoundland and Labrador specified that all electronic data for this study shall be stored on a secure Health Science Information and Media Service (HSIMS) server with restricted password-protected access as well as password-protected databases, and the data guardian is Dr. Guangju Zhai. The contact information for HREA of Newfoundland and Labrador is as follows: Email: info@hrea.ca <mailto:info@hrea.ca>, Phone: (709) 864-8871."

Before we proceed, we’ll require some additional information to ensure your submission adheres to the PLOS ONE Data Availability policy (https://journals.plos.org/plosone/s/data-availability <https://journals.plos.org/plosone/s/data-availability>)

PLOS asks that authors provide a statement which explains where and how a manuscript's minimal data set can be found. The minimal data set is defined as the data set used to reach the conclusions drawn in the manuscript with related metadata and methods, and any additional data required to replicate the reported study findings in their entirety. This may include: a.) The values behind the means, standard deviations and other measures reported; b.) The values used to build graphs; c.) The points extracted from images for analysis (https://journals.plos.org/plosone/s/data-availability#loc-minimal-data-set-definition <https://journals.plos.org/plosone/s/data-availability#loc-minimal-data-set-definition>).

If a manuscript's minimal data set cannot be made publicly available, PLOS ONE requires that authors provide a Data Availability statement with specific information to facilitate data access requests. For data that is subject to restrictions, we ask that authors include the following information in their statement:

-The reason for the restriction (i.e., confidentiality of patients, etc.) -The name of the restricting institution -A non-author, institutional point of contact (preferably email) for a data access committee, ethics committee, or other institutional body that other researchers would require to request access to your data. Note that it is not acceptable for an author to be the sole named individual responsible for ensuring data access.

We note that you have provided what appears to be a Data Availability statement associated with restricted data via your Response to Reviewers, but have not updated your statement in the submission system. A manuscript's Data Availability statement should only appear in the submission system, and is generally not included in the body of a manuscript. In your response to this message, please reply to the following queries:

-Can researchers contact the HREA of Newfoundland and Labrador to request access to data via the email address you provided (info@hrea.ca <mailto:info@hrea.ca>)?

-Please provide a revised Data Availability statement which includes the information conveyed via your Response to Reviewers. Please ensure that it includes all three pieces of information requested above.

Response:

We have now removed the Data Availability Statement from our manuscript (changed tracked and clean versions).

In the submission system, we have provided a revised Data Availability Statement which includes the information conveyed via our Response to Reviewers, and it includes information on 1) The reason for the restriction (ethics application approved); 2) The name of the restricting institution (the Health Research Ethics Authority (HREA) of Newfoundland and Labrador); 3) A non-author, institutional point of contact (and email) for an ethics committee (the HREA of Newfoundland and Labrador, info@hrea.ca) that other researchers would require to request access to our data (in addition to the corresponding author). Researchers can contact the HREA of Newfoundland and Labrador to request access to data via the email address we provided (info@hrea.ca).

Please see our response to requirement #4 above.

Reviewers' comments:

Reviewer's Responses to Questions

Comments to the Author

1. Is the manuscript technically sound, and do the data support the conclusions?

Reviewer #1: Partly

Reviewer #2: Partly

Reviewer #3: No

2. Has the statistical analysis been performed appropriately and rigorously?

Reviewer #1: Yes

Reviewer #2: No

Reviewer #3: I Don't Know

3. Have the authors made all data underlying the findings in their manuscript fully available?

Reviewer #1: No

Reviewer #2: No

Reviewer #3: Yes

4. Is the manuscript presented in an intelligible fashion and written in standard English?

Reviewer #1: Yes

Reviewer #2: Yes

Reviewer #3: Yes

5. Review Comments to the Author

Reviewer #1:

The manuscript “Postoperative joint pain is associated with long-term all-cause mortality after total joint Arthroplasty” examined the effect of post-operative pain on mortality among patients undergoing TJR. Specific comments and clarifications are below:

1. Abstract (lines 14-16): “… their postoperative joint pain at least one-year after TJA…” Was WOMAC collected at several time points after surgery? Please clarify.

Response:

The postoperative joint pain was accessed only once for each patient. We have now clarified this in the Abstract (Revised Manuscript with Track Changes, Page 2, Line 19) and Methods (Revised Manuscript with Track Changes, Page 6, Line 85) sections.

2. Methods, Page 5: Please include the exact duration of the follow-up period (e.g., last date, average # of years of follow-up) and the number of post-operative measurements.

Response:

The average follow-up time for postoperative joint pain assessment and the average time from TJA to mortality data extraction were already specified in the Results section (4.0 and 11.0 years after TJA for postoperative joint pain assessment and mortality data extraction, respectively, Manuscript with Track Changes, page 11, line 151, line 154), and we have now added the last follow-up date for postoperative joint pain assessment (July 29, 2019. Manuscript with Track Changes, page 11, lines 151-152) as well as the date of mortality data extraction (June 3, 2024 and January 15, 2025. Manuscript with Track Changes, page 11, lines 154-156). We have also specified in the Methods section that data on postoperative joint pain evaluated by the Western Ontario and McMaster Universities Osteoarthritis Index (WOMAC) Likert 3.0 pain subscale (five questions, Manuscript with Track Changes, page 6, lines 85-89) and three mortality variables were collected (Manuscript with Track Changes, page 6, lines 104-105) after the TJA.

3. Page 5, lines 79-80: Please clarify what post-operative time point was used for the analysis.

Response:

Please see response to comment #2.

4. Page 6: I think that the label ‘sustained pain’ is misleading as it is based on a one-time pain reporting based on a question asking for pain in the past 48 hours.

Furthermore, I think that post-operative pain status is not the best variable to accomplish the study objectives, as it does not account for the level of pre-operative pain. I think a variable looking at changes in pain pre- and post- operatively is more appropriate for the study question. Please comment on the choice of the pain variables used.

Response:

We thank the reviewer for pointing this out. The term “sustained pain” was to describe the status of having j

---

## [Decision Letter · Decision Letter 1]

PONE-D-25-06222R1Postoperative joint pain is associated with long-term all-cause mortality after total joint arthroplastyPLOS ONE

Dear Dr. Zhai,

Thank you for submitting your manuscript to PLOS ONE. After careful consideration, we feel that it has merit but does not fully meet PLOS ONE’s publication criteria as it currently stands. Therefore, we invite you to submit a revised version of the manuscript that addresses the points raised during the review process.

We look forward to receiving your revised manuscript.

Kind regards,

Yuanyuan Wang, PhD

Academic Editor

PLOS ONE

Journal Requirements:

**Additional Editor Comments:**

The authors have addressed the reviewers' comments appropriately. One reviewer requested a minor revision by adding a limitation.

Reviewers' comments:

Reviewer's Responses to Questions

**Comments to the Author**

1. If the authors have adequately addressed your comments raised in a previous round of review and you feel that this manuscript is now acceptable for publication, you may indicate that here to bypass the “Comments to the Author” section, enter your conflict of interest statement in the “Confidential to Editor” section, and submit your "Accept" recommendation.

Reviewer #1: All comments have been addressed

2. Is the manuscript technically sound, and do the data support the conclusions?

Reviewer #1: Yes

3. Has the statistical analysis been performed appropriately and rigorously? 

Reviewer #1: Yes

4. Have the authors made all data underlying the findings in their manuscript fully available?

Reviewer #1: Yes

5. Is the manuscript presented in an intelligible fashion and written in standard English?

Reviewer #1: Yes

6. Review Comments to the Author

Reviewer #1: I thank the authors for addressing my comments thoroughly. My only remaining comment, though, is that I suggest adding a sentence to the limitations section pointing out that the analysis did not account for pre-operative level of pain.

7. PLOS authors have the option to publish the peer review history of their article (what does this mean? ). If published, this will include your full peer review and any attached files.

**Do you want your identity to be public for this peer review?** For information about this choice, including consent withdrawal, please see our Privacy Policy .

Reviewer #1: No

---

## [Author Response · Author response to Decision Letter 2]

18 Jun 2025

Journal Requirements:

Response:

We have reviewed the reference list and corrected reference #27. In PubMed, the journal name for this article is listed as “Arthritis Rheum”, which was presented as the journal name for this reference when we added it using PMID number. However, the full text shows that this article was published in “Arthritis Care & Research”. The volume, issue, and page numbers were correct. We have corrected the journal name of this reference. We did not cite any papers that have been retracted. The reference list is now complete and correct.

Additional Editor Comments:

The authors have addressed the reviewers' comments appropriately. One reviewer requested a minor revision by adding a limitation.

Response:

We thank the editor and reviewers for approving the changes we have made according to the editor and reviewers’ constructive comments. We have added a limitation as suggested by reviewer #1.

Reviewers' comments:

Reviewer's Responses to Questions

Comments to the Author

1. If the authors have adequately addressed your comments raised in a previous round of review and you feel that this manuscript is now acceptable for publication, you may indicate that here to bypass the “Comments to the Author” section, enter your conflict of interest statement in the “Confidential to Editor” section, and submit your "Accept" recommendation.

Reviewer #1: All comments have been addressed

2. Is the manuscript technically sound, and do the data support the conclusions?

Reviewer #1: Yes

3. Has the statistical analysis been performed appropriately and rigorously?

Reviewer #1: Yes

4. Have the authors made all data underlying the findings in their manuscript fully available?

Reviewer #1: Yes

5. Is the manuscript presented in an intelligible fashion and written in standard English?

Reviewer #1: Yes

6. Review Comments to the Author

Reviewer #1: I thank the authors for addressing my comments thoroughly. My only remaining comment, though, is that I suggest adding a sentence to the limitations section pointing out that the analysis did not account for pre-operative level of pain.

Response:

We thank the reviewer for the suggestion. We have added this limitation in the revised manuscript (Revised Manuscript with Track Changes, Page 24, Lines 311-315).

7. PLOS authors have the option to publish the peer review history of their article (what does this mean?). If published, this will include your full peer review and any attached files.

Do you want your identity to be public for this peer review? For information about this choice, including consent withdrawal, please see our Privacy Policy.

Reviewer #1: No

---

## [Editor Report · Decision Letter 2]

Postoperative joint pain is associated with long-term all-cause mortality after total joint arthroplasty

PONE-D-25-06222R2

Dear Dr. Zhai,

We’re pleased to inform you that your manuscript has been judged scientifically suitable for publication and will be formally accepted for publication once it meets all outstanding technical requirements.

Kind regards,

Yuanyuan Wang, PhD

Academic Editor

PLOS ONE

Additional Editor Comments (optional):

The authors have addressed all the reviewers' comments.
---

## [Editor Report · Acceptance letter]

PONE-D-25-06222R2

PLOS ONE

Dear Dr. Zhai,

I'm pleased to inform you that your manuscript has been deemed suitable for publication in PLOS ONE. Congratulations! Your manuscript is now being handed over to our production team.

Kind regards,

on behalf of

Dr. Yuanyuan Wang

Academic Editor

PLOS ONE